# The effects of substance use on non-communicable diseases among older adults aged 60 and above in the North-eastern States of India

**Sasanka Boro**⊙\*, **Nandita Saikia**

Department of Public Health and Mortality Studies, International Institute for Population Sciences, Mumbai, India

\* boro485@gmail.com

## Abstract

### Introduction

The North-eastern region of India has a relatively higher prevalence of substance use, which together with poor dietary practices and a lack of physical activity is one of the key risk factors for NCDs among older adults in the region. Understanding the prevalence of NCDs and their relationship to substance use can help develop preventive strategies and sensitization in North-eastern India.

### Objective

To assess the prevalence of NCDs and the strength of the association of substance abuse among the geriatric population of North-eastern states in India, for the development of preventive strategies.

### Methods

Data from the Longitudinal Ageing Study in India (LASI Wave-I, 2017–18) were drawn to develop this paper. The bi-variate and binary logistic regression analyses were carried out to predict the association between non-communicable diseases and substance use adjusting select socio-demographic characteristics.

### Results

The paper revealed the prevalence of NCDs among urban people (61.45%) is higher than among rural people (42.45%). Hypertension (37.29%) can be seen as the most prevalent disease among the following given NCDs followed by Diabetes (8.94%). The chances of having Cancer are nineteen times higher (OR = 19.8; C.I. = 18.82–20.83) if an individual has past smoking behaviour after controlling for socio-demographic and physical activity variables.

**Data Availability Statement:** The data underlying the results presented in the study are available from (GATEWAY TO GLOBAL AGING DATA, https://g2aging.org/?section=lasi-downloads).

**Funding:** The authors received no specific funding for this work.

**Competing interests:** NO authors have competing interests.

**Abbreviations:** COTPA, Cigarettes and other Tobacco Products Act; CVD, Cardiovascular Diseases; DHS, Demographic and Health Surveys; FCTC, Framework Convention on Tobacco Control; GATS, Global Adult Tobacco Survey; LASI, Longitudinal Aging Survey of India; MDG, Millennium Development Goals; MPCE, Monthly per Capita Expenditure; NCDs, Non-Communicable Diseases; NE, North East; NFHS, National Family and Health Survey; SDG, Sustainable Development Goals; STATA, Statistics and Data; UN, United Nations; UNDP, United nations Development Programme; WHO, World Health Organisation.

## Conclusion

Since, the high prevalence of hypertension correlated with the high level of substance abuse, require immediate attention to develop appropriate intervention strategies for its control (substance abuse) and prevention of hypertension. In a lower middle-income country like India, preventive measures, rather than curative measures will be cost-effective and helpful.

## Introduction

Epidemiological transition theory describes how the mortality and morbidity of a society change over time due to changes in social and economic conditions. It involves a shift from a predominance of communicable diseases as the leading causes of death to a predominance of non-communicable diseases, such as cardiovascular disease, cancer, and diabetes, typically associated with older age [1]. While the transition is driven by a range of social, economic, and environmental factors, demographic factors such as population ageing are key drivers. Population ageing demonstrates the shift in the age structure of a population over time due to fertility and mortality decline [2]. This shift is characterized by an increase in the proportion of people aged 60 and over, and a decrease in the proportion of young people. This changing age structure of a population has implications for health and wellbeing, particularly with regard to the increased burden of chronic diseases in an ageing population [3].

India along with other nations is going through unprecedented demographic changes, increased life expectancy and fertility decline have changed the age structure of India's Older adults population, which is rapidly growing and emerging as a serious source of concern for the government and policymakers [4]. It not only gives extra years to survive but also serious health problems, as health conditions are more likely to occur in older age. The emergence of NCDs can be related to the population's increased lifespan and are the most common, particularly among Older adults [5]. Thus, the goal should not be limited only to adding an extra number of years but to age healthily [6], healthy ageing is important in order to prevent the onset of NCDs and improve quality of life as people age [7]. As people age, their risk for developing NCDs increases, which can lead to reduced quality of life [8]. NCDs can also lead to physical and mental health problems, such as increased pain, fatigue, and depression [9]. Furthermore, unhealthy lifestyles, such as substance use, lack of physical activity, and unhealthy diets, can contribute to the development of NCDs and poor quality of life [10]. Therefore, it is essential to maintain healthy lifestyle habits in order to reduce the risk of NCDs and improve the quality of life.

Substance use is defined as the consumption of any psychoactive substance, such as tobacco, alcohol, or drug (legal or illicit drugs) other than those for medical use [11]. Previous literature found that substance consumption is one of the main factors for the burden of disease, disability, premature death, and non-communicable diseases (NCDs) among older adults. In most countries, it is a significant public health issue and is a challenging element of the overall healthcare system [12, 13]. In India, Alcohol (21.4%) was the primary substance used (apart from tobacco) followed by cannabis (3.0%) and opioids (0.7%) [14]. Numerous studies have been conducted to date that show high rates of tobacco and alcohol consumption, but no other study has attempted to link substance use to the occurrence of NCDs, Therefore, this study aims to investigate tobacco and alcohol use among older adults in north-east India. Specifically, the objective of this paper is to examine the associations of various non-

communicable diseases with substance use, i.e., (1) NCDs and Ever smoked/smokeless tobacco use (2) NCDs and Ever-consumed Alcohol (3) NCDs and Current Smokers (4) NCDs and Current use smokeless tobacco products (5) NCDs and Current alcohol consumption.

## Substance use as a risk factor for non-communicable diseases

Noncommunicable diseases (NCDs) include a wide range of diseases like cardiovascular diseases, cancer, diabetes, chronic respiratory diseases and neurological or psychiatric diseases which affect 41 million people annually and account for 71% of all deaths globally. Every year, more than 15 million people between the ages of 30 and 69 die due to an NCD. The low- and middle-income countries account for 85% of these "premature" deaths while they also account for 77% of all NCD deaths.

According to one study, tobacco use alone is responsible for one-sixth of all NCD-related deaths, while alcohol use accounts for more than two-thirds of all new cases of NCDs and raises the risk of complications in those who already have them, because of nicotine dependence, more than a billion people smoke or chew tobacco every day, and 15,000 people pass away from diseases associated with tobacco use [15].

Tobacco use, physical inactivity, harmful alcohol use, and poor dietary habits are the main risk factors of dying from an NCD [16]. India accounts for 5.87 million (60%) of all NCD-related fatalities, which accounts for more than two-thirds of all NCD-related deaths in the WHO's South East Asia region. [17].

GATS [18] estimates that 266.8 million adults in India, who are 15 years of age and older, use tobacco in some way on a daily basis which accounts for 28.6% of them. In India, every tenth adult smokes tobacco, which amounts to 99.5 million people, and every fifth adult uses smokeless tobacco, which amounts to 199.4 million people, with the North-eastern area of the country displaying the highest prevalence of all the other states. [18]. Additionally, according to estimates from the World Health Organization (WHO), adult alcohol per capita consumption (APC) in India has significantly climbed from 2.4 liters, 4.3 liters, and 5.7 liters in 2005, 2010, and 2016 correspondingly [19].

## Northeast India

Assam, Arunachal Pradesh, Manipur, Meghalaya, Mizoram, Nagaland, Sikkim, and Tripura are the eight Indian states that represent Northeast India. Together, they are home to 3.8% of the nation's total population, and of that, around 81.6% reside in rural areas [20].

A few small-scale studies found that the prevalence of substance use is higher in the North-eastern states of India [21–23]. According to a study conducted on the Mishing tribe in Assam, nearly half of the women consume alcohol, which is higher than the reported rate for women overall, substance use was very high in this population, with more than 60% of the population using alcohol and tobacco [24]. A national study shows the current tobacco usage among older adults with at least one chronic disease is significantly higher in the Northeast region (47.6%) in India [25].

According to a pilot study executed in Assam and Meghalaya, the prevalence of substance use in Assam and Meghalaya both was 29.4% tobacco (20.5% chewers and 12.7% smokers), 12.5% alcohol, and 4.9% opium. In Meghalaya, the prevalence of tobacco use was high (41.7%), mostly because there were so many male smokers and female chewers [26]. According to research by [27], more than half of adults in the states of Meghalaya, Manipur, Tripura, and Mizoram reported using tobacco at some point between 2009–2010 and 2012–2013. Meghalaya and Manipur also showed an increase in tobacco usage among women.

Studies have revealed that the North-eastern region of India has a lower overall prevalence of non-communicable diseases than the other regions of the country, however, the prevalence of CVD and Diabetes were high as compared with the central regions [28]. As substance consumption are leading behavioural and lifestyle risk factor for noncommunicable diseases all over the world, it is important to study how the increase in alcohol and tobacco consumption is influencing the burden of NCDs in the North-eastern states. Previous literature shows substance use and its association with noncommunicable diseases among adults and youth aged 15–54, however, there has not been any literature specifically on Older adults in the North-eastern region [23, 26, 29, 30]. This study includes age group from 60 and above i.e., older adults, because non-communicable diseases are most prevalent and common in these stages and their previous lifestyle behaviour comes into life by affecting the occurrence of these diseases.

## Data and methodology

**Data source.** The study uses the Longitudinal Ageing Study in India (LASI) wave-1 Individual dataset to assess the prevalence and association of substance use and noncommunicable diseases among older adults aged 45 and above in the North-eastern states of India. The goal of LASI was to study a 45-year-old nationwide sample of India's population. In order to give policymakers, the knowledge they need to enhance health and health behaviours in this demographic segment, the project is gathering information on the social, economic, and health circumstances of older people throughout India. Every two years for the next 25 years, a representative sample of the population aged 45 and older will be followed, together with their spouses, regardless of age, with refreshment samples to account for attrition due to death, dislocation, non-contact, and refusal. In Wave 1 of the LASI, 72,250 adults aged 45 and older and their spouses were included, including 31,464 older adults aged 60 and older and 6,749 oldest-old individuals aged 75 and older from 35 Indian states and union territories.

## Socioeconomic and demographic variables

Socio-economic and Demographic variables namely Sex (Male and Female), Age Group (60–69, 70–79, and 80 and above), Place of residence (Urban and Rural), Caste (SC, ST, OBC, and Others), Religion (Hindu, Muslim, Christian, Buddhist, and Others), MPCE Quintile (Lowest, Lower, Middle, Higher, Highest), The assessment of monthly per capita consumption expenditure (MPCE) quintile was conducted utilizing household consumption data. Two sets of questions, comprising 11 for food and 29 for non-food items, were administered to sample households. Both food and non-food expenditures were standardized over a 30-day reference period. The MPCE was computed as the comprehensive measure of consumption. Subsequently, this variable was divided into five quintiles, ranging from the poorest to the richest. No. of years of education (No education, Less than 5 years of education, 5–9 years of education and 10 or more), Marital status (Currently married, Widowed, and Others), Work status (Currently working, Worked before but not currently working and never worked) were selected based on previous literature reviews.

## Substance consumption and physical activity variables

We analyzed five dichotomous indicators of substance use viz. ever smoked/ used smokeless tobacco, ever consumed alcohol, currently consuming smokeless tobacco, currently consuming alcohol, currently smoking, and physical activity variable viz engaged in vigorous activities which are categorised into 5 categories (Every Day, more than once a week, once a week, one or three times a month, Hardly or never).

### Dependent variable

In the process of data collection, information regarding 12 different self-reported chronic physical diseases and conditions (including hypertension, diabetes, cancer, chronic lung disease, chronic heart disease, stroke, arthritis, osteoporosis or other bone/joint diseases, neurological or psychiatric problems, hypercholesterolemia, thyroid disease, gastrointestinal problems, and chronic renal disease) was gathered and were coded as either "Yes" or "No" and was consequently used in our study.

### Statistical analysis

Bivariate analyses are done to calculate the prevalence of those who have chronic diseases namely Hypertension, diabetes, cancer, lung disease, heart disease, high cholesterol, bone or joint disease, stroke and neurological/psychiatric diseases and lastly having any one of the following diseases by their socio-economic and demographic characteristics.

Since the dependent variables, i.e. NCDs are binary in nature, with only two categories–"Yes" and "No" and when the outcome variable is dichotomous in nature then binary logistic regression is the most commonly used model [31]. Thus, Binary logistic regression is used to identify significant relationships between ever-consumed substances and non-communicable diseases after controlling for the socio-economic and demographic variables among older adults aged 60 and above in the North-eastern states of India. The equation can be expressed as follows:

$$\log\left(\frac{p}{1-p}\right) = \beta_0 + \beta_1 x_1 + \beta_2 x_2 + ... + \beta_n x_n$$

Where p is the probability of occurrence of non-communicable disease, log is the natural logarithm, $\beta_0$ is the intercept, $\beta_1, \beta_2, \ldots, \beta_n$ are the coefficients associated with the predictor variables, $x_1 x_2, \ldots, x_n$.

## Results

### Sample description

Table 1 presents the descriptive statistics of the study population in addition to the characteristics of the Indian population to compare. Our sample consisted of 4155 individuals who were from the North-eastern states of India. Overall, the share of study participants was highest in the 60–69 age group, and males (52%) outnumbered females (48%). Most of the study population resided in rural areas (76%) and the majority of the study population belonged to the Christian religion (46%), followed by Hindus (39%). Only 11% of the participants had completed 10 or more years of education. 62% of the sample were currently married. More than half of the study participants were not engaged in work or had never worked before in the formal sector.

### Prevalence of non-communicable diseases by background characteristics

Fig 1 shows the state wise prevalence of non-communicable diseases among the older adults of North-eastern states of India, it can see that Sikkim has the highest share of people suffering from any NCD(65.89%) followed by Tripura(51.25%), Assam(48%), Manipur (46.72%), Mizoram (46.48%), Meghalaya (39.28%), Arunachal Pradesh (37.46%) and Nagaland (21.29%).

It was observed that people suffering from various diseases have a previous history of tobacco consumption (smoking or smokeless tobacco) as follows: hypertension (48%), diabetes (42%), cancer (56%), lung diseases (69%), heart diseases (52%), high cholesterol (36%),

**Table 1. Description of study population.**

| Background Characteristics | | Northeast | | India | |
|---|---|---|---|---|---|
| | | Frequency (n) | Percentage (%) | Frequency (n) | Percentage (%) |
| Age Group | 60–69 | 2,372 | 57.09 | 19,211 | 60.22 |
| | 70–79 | 1,226 | 29.51 | 9,250 | 29 |
| | 80+ | 557 | 13.41 | 3,441 | 10.79 |
| Sex | Male | 2,004 | 48.23 | 15,340 | 48.08 |
| | Female | 2,151 | 51.77 | 16,562 | 51.92 |
| Place of residence | Rural | 3,145 | 75.69 | 21,085 | 66.09 |
| | Urban | 1,010 | 24.31 | 10,817 | 33.91 |
| Caste | SC | 301 | 7.24 | 5,157 | 16.17 |
| | ST | 2,408 | 57.95 | 5,334 | 16.72 |
| | OBC | 787 | 18.94 | 12,137 | 38.04 |
| | Others | 659 | 15.86 | 9,274 | 29.07 |
| Religion | Hindu | 1,609 | 38.72 | 23,292 | 73.01 |
| | Muslim | 219 | 5.27 | 3,731 | 11.7 |
| | Christian | 1,944 | 46.79 | 3,194 | 10.01 |
| | Others | 1,609 | 9.22 | 1,685 | 5.28 |
| MPCE quintile | Lower | 738 | 17.76 | 6,580 | 20.63 |
| | Lowest | 834 | 20.07 | 6,573 | 20.6 |
| | Middle | 876 | 21.08 | 6,502 | 20.38 |
| | Higher | 876 | 21.08 | 6,259 | 19.62 |
| | Highest | 831 | 20.00 | 5,988 | 18.77 |
| Education | No Education | 2,167 | 52.15 | 17,190 | 53.89 |
| | Less than 5 years | 720 | 17.33 | 3,835 | 12.02 |
| | 5–9 years | 804 | 19.35 | 6,065 | 19.01 |
| | 10 or more years | 464 | 11.17 | 4,811 | 15.08 |
| Marital Status | Currently Married | 2,587 | 62.26 | 20,212 | 63.36 |
| | Widowed | 1,417 | 34.10 | 10,845 | 33.99 |
| | Others | 151 | 3.63 | 845 | 2.65 |
| Work Status | Currently working | 1,442 | 34.71 | 9,373 | 29.39 |
| | Worked before | 1,494 | 35.96 | 13,476 | 42.26 |
| | Never worked | 1,219 | 29.34 | 9,038 | 28.34 |
| Total | | 4155 | 100 | 31,887 | 100 |

stroke (54%), bone or joint diseases (48%), and neurological or psychiatric diseases (64%) (Fig 2). Additionally, the history of alcohol consumption among these individuals is: hypertension (21%), diabetes (23%), cancer (17%), lung diseases (27%), heart diseases (18%), high cholesterol (20%), stroke (37%), bone or joint diseases (20%), and neurological or psychiatric diseases (19%) (Fig 3).

Table 2 shows the prevalence of non-communicable disease by background characteristics and by state for people aged 60 and above in the North-eastern states and of India. Prevalence of all the NCDs among the urban people is higher than the rural people except cancer where it is more prevalent in the rural areas. Hypertension (37%) can be seen as the most prevalent disease among the following given NCDs followed by Diabetes (9%) then Bone or Joint Disease (7%) among the older adults of North-eastern states of India. ST older adults have less proportion suffering from Hypertension (29.04%), Diabetes (7.32%), lung diseases (3.56%), heart disease (1.83%), and Bone/Joint Disease (4.75%) compared with the other categories. The

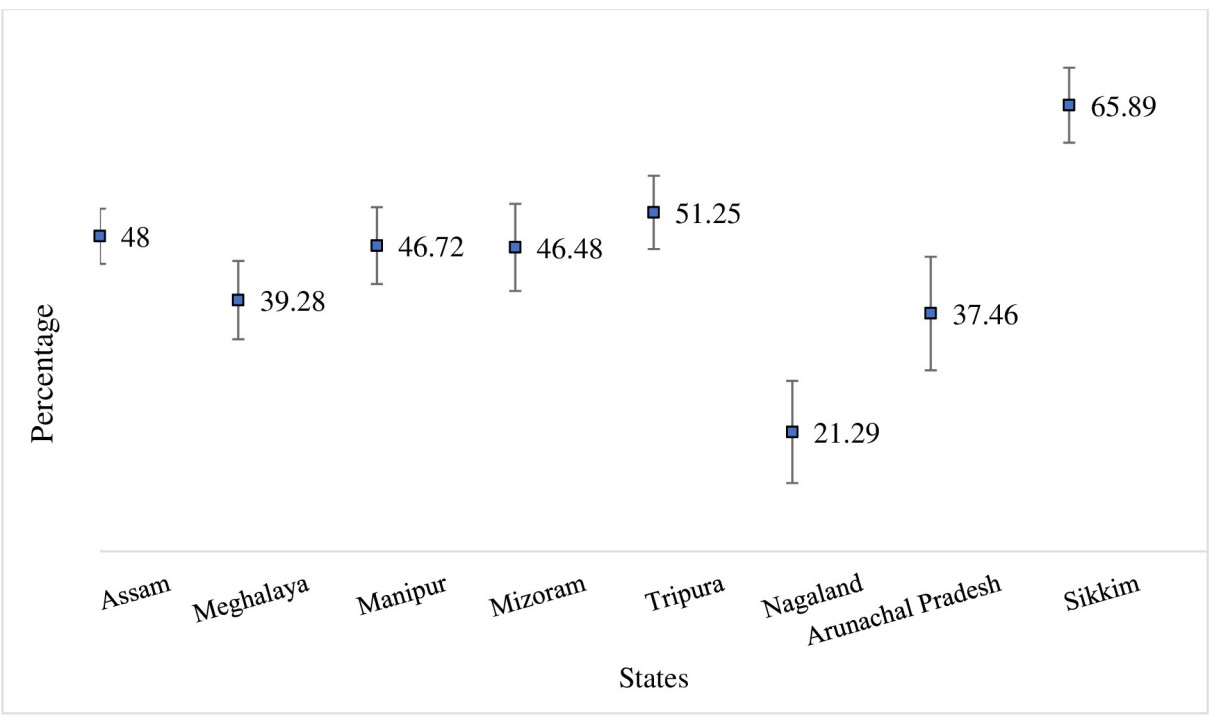

**Fig 1. State wise prevalence of older adults aged 60 and above suffering from NCDs in the North-Eastern states of India, 2017–18.**

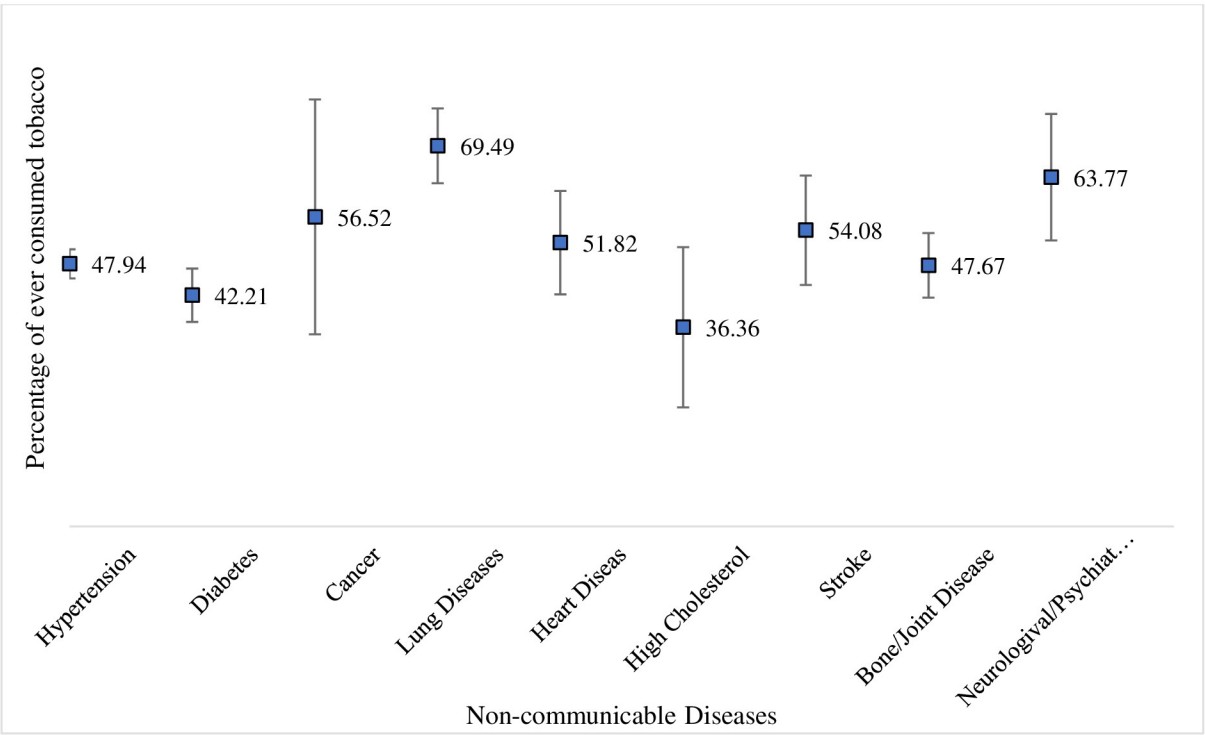

**Fig 2. Proportion of older adults aged 60 and above suffering from NCDs with past tobacco consumption behaviour.**

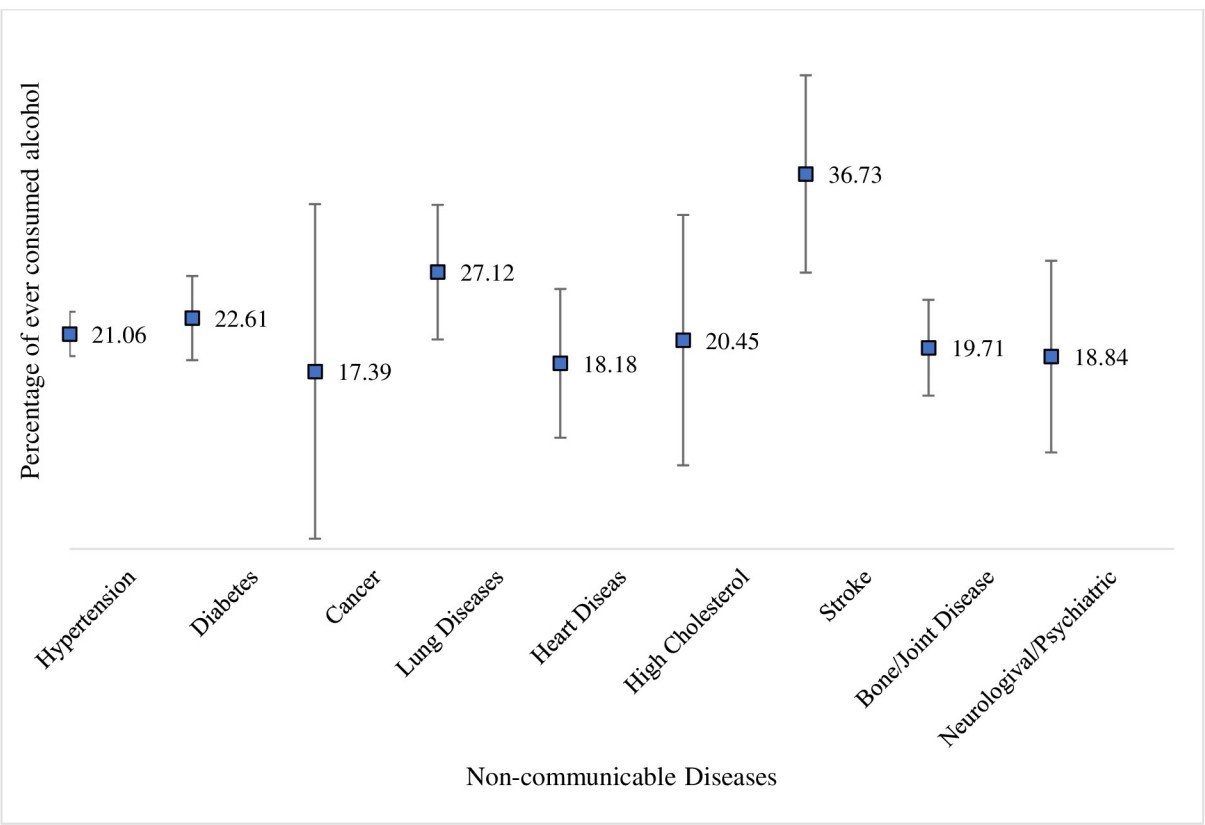

**Fig 3. Proportion of older adults aged 60 and above suffering from NCDs with past alcohol consumption behaviour.**

prevalence of the diseases for currently working people was less than for the people who were not working currently. It was observed that there was no clear pattern in the effect of education on the prevalence of all non-communicable diseases. However, our study found that individuals in the highest educational attainment category experienced greater levels of suffering compared to those with fewer years of education.

Hypertension, Cancer, heart diseases, high cholesterol, bone or Joint Diseases and Neurological/Psychiatric Diseases were high among the older women than men.

Hypertension (54.54%), Diabetes (17.15%) and heart diseases (5.4%) were found to be highest in the state of Sikkim. Cancer is the highest in the State of Mizoram (1.17%). Lung Disease is the highest in the state of Tripura (9.35%). Whereas Neuro-Psychiatric Diseases can be seen highest in the state of Mizoram (2.35%).

## Association between substance use and non-communicable diseases

Table 3 shows the binary logistic regression results of noncommunicable diseases with ever smoked or used tobacco products, ever consumed alcohol, currently smoking people, currently consuming tobacco products and currently consuming alcohol which is adjusted for physical activity, socioeconomic and demographic variables.

Ever Consumption of alcohol causes 1.33, 2.15 1.12, 1.21, 1.7, and 1.4 times more risk of suffering from Hypertension, Diabetes, Heart Diseases, Bone/Joint Disease, Stroke and Neuro/ Psychiatric Diseases respectively as compared with individuals who have never consumed alcohol. Ever consumed Smoking and tobacco consumption creates 2.31 and 2.91 times more

**Table 2. Prevalence of non-communicable diseases by background characteristics among older adults aged 60 and above in the North-eastern states in India, 2017–18.**

| Background Characteristics | | Hypertension | Diabetes | Cancer | Lung Disease | Heart Disease | High Cholesterol | Bone/Joint Disease | Stroke | Neuro/Psychiatric Disease | Any NCD |
|---|---|---|---|---|---|---|---|---|---|---|---|
| **Sex** | **Male** | 34.55 | 11.39 | 0.41 | 6.1 | 2.86 | 0.89 | 6.41 | 3.96 | 1.69 | 44.15 |
| | **Female** | 39.63 | 6.85 | 0.57 | 3.81 | 4.25 | 1.08 | 7.49 | 1.71 | 2.69 | 48.27 |
| **Age Group** | **60–69** | 34.02 | 9.13 | 0.66 | 3.87 | 3.47 | 1.23 | 6.51 | 2.63 | 2.24 | 43.06 |
| | **60–75** | 39.98 | 8.77 | 0.27 | 5.29 | 3.87 | 0.66 | 7.98 | 2.61 | 2.35 | 49.42 |
| | **75+** | 47.68 | 8.37 | 0.22 | 9.1 | 3.65 | 0.59 | 6.9 | 3.73 | 1.89 | 55.99 |
| **Place of residence** | **Rural** | 34.24 | 7.1 | 0.54 | 4.75 | 2.7 | 0.32 | 6.27 | 2.03 | 1.73 | 42.45 |
| | **Urban** | 49.04 | 16.03 | 0.34 | 5.34 | 7.1 | 3.57 | 9.76 | 5.51 | 4.18 | 61.45 |
| **Caste** | **SC** | 35.76 | 12.13 | 0.72 | 7.91 | 9.26 | 0.17 | 5.69 | 0.65 | 2.44 | 47.33 |
| | **ST** | 29.04 | 7.32 | 0.48 | 3.56 | 1.83 | 0.91 | 4.75 | 2.73 | 1.64 | 36.98 |
| | **OBC** | 42.81 | 7.93 | 0.04 | 5.7 | 2.72 | 0.43 | 8.63 | 2.69 | 2.28 | 50.91 |
| | **Others** | 41.13 | 10.68 | 0.92 | 4.42 | 4.58 | 1.94 | 8.18 | 3.54 | 2.76 | 51.63 |
| **Religion** | **Hindu** | 40.27 | 9.49 | 0.39 | 5.58 | 4.12 | 1.08 | 6.93 | 2.8 | 2.29 | 49.13 |
| | **Muslim** | 36.98 | 8.32 | 0.55 | 3.08 | 5.54 | 0 | 6.68 | 3.37 | 1.23 | 45.84 |
| | **Christian** | 28.22 | 7.57 | 0.63 | 3.57 | 0.98 | 1.17 | 7.14 | 1.94 | 3.08 | 37.52 |
| | **Others** | 30.77 | 8.23 | 1.4 | 5.16 | 0.69 | 1.96 | 8.21 | 3.38 | 0.99 | 42.99 |
| **MPCE quintile** | **Lowest** | 28.79 | 4.19 | 0 | 4.99 | 2.19 | 0.76 | 2.78 | 0.87 | 0.42 | 34.49 |
| | **Lower** | 35.07 | 5.52 | 0.19 | 3.19 | 2.73 | 0.26 | 5.86 | 3.51 | 0.52 | 44.85 |
| | **Middle** | 41.65 | 12.28 | 0.38 | 6.04 | 4.21 | 0.61 | 5.24 | 2.37 | 4.65 | 51.61 |
| | **Higher** | 40.3 | 10.81 | 0.03 | 5.84 | 3.83 | 0.85 | 8.81 | 2.68 | 2.27 | 49.23 |
| | **Highest** | 41.62 | 12.94 | 2.1 | 4.53 | 5.45 | 2.81 | 13.34 | 4.42 | 3.71 | 52.75 |
| **Education** | **No Education** | 31.91 | 4.52 | 0.27 | 5.04 | 2.67 | 0.31 | 6.72 | 2.19 | 1.86 | 40.96 |
| | **Less than 5 years** | 40.23 | 11.06 | 0.15 | 4.14 | 4.52 | 0.16 | 4.86 | 1.51 | 1.36 | 47.51 |
| | **5–9 years** | 39.78 | 9.16 | 0.92 | 5 | 4.37 | 1 | 6.94 | 2.78 | 2.92 | 48.76 |
| | **10 or more years** | 48.92 | 21.77 | 1.05 | 4.91 | 4.65 | 4.39 | 10.59 | | 3.51 | 60.4 |
| **Marital Status** | **Currently Married** | 34.76 | 10.91 | 0.58 | 5.76 | 3.55 | 1.46 | 7.6 | 3.05 | 1.33 | 45.31 |
| | **Widowed** | 41.26 | 6.14 | 0.41 | 3.81 | 3.74 | 0.15 | 6.28 | 2.43 | 3.59 | 48.54 |
| | **Others** | 31.8 | 7.29 | 0 | 0 | 2.74 | 4.15 | 3.22 | 0.23 | 1.32 | 33.99 |
| **Work Status** | **Currently working** | 27.33 | 6.5 | 0.06 | 3.29 | 1.88 | 0.72 | 5.71 | 1.47 | 0.65 | 34.19 |
| | **Worked before** | 41.91 | 13.19 | 1.02 | 6.88 | 5.13 | 1.66 | 7.82 | 4.15 | 2.44 | 53.71 |
| | **Never worked** | 41.39 | 6.14 | 0.29 | 3.96 | 3.45 | 0.44 | 7.23 | 2.29 | 3.53 | 49.31 |
| **State** | **Assam** | 40.14 | 8.58 | 0.4 | 4.71 | 4.2 | 0.49 | 6.42 | 2.75 | 2.18 | 48 |
| | **Meghalaya** | 35.6 | 5.52 | 0.19 | 1.96 | 1.39 | 1.05 | 4.39 | 0.76 | 1.42 | 39.28 |
| | **Manipur** | 33.47 | 11.85 | 1.11 | 3.76 | 4.08 | 5.4 | 6.2 | 3.66 | 1.6 | 46.72 |
| | **Mizoram** | 31.28 | 9.18 | 1.17 | 8.94 | 1.17 | 0.85 | 9.08 | 3.33 | 3.29 | 46.48 |
| | **Tripura** | 36.67 | 10.65 | 0.7 | 9.35 | 3.81 | 0.87 | 10.76 | 3.95 | 2.52 | 51.25 |
| | **Nagaland** | 15.1 | 6.8 | 0.08 | 1.02 | 0.37 | 0.37 | 3.61 | 1.39 | 5.33 | 21.29 |
| | **Arunachal Pradesh** | 26.64 | 8.51 | 0.56 | 4.37 | 0.29 | 0 | 13.16 | 1.96 | 1.16 | 37.46 |
| | **Sikkim** | 54.54 | 17.15 | 0 | 1.65 | 5.4 | 0.8 | 11.32 | 0.36 | 1.47 | 65.89 |
| **Total** | | 37.29 | 8.94 | 0.5 | 4.87 | 3.61 | 0.99 | 6.99 | 2.75 | 2.23 | 46.37 |

**Table 3. Binary logistic regression output table of non-communicable diseases by substance consuming behaviour among older adults aged 60 and above in the North-eastern states of India, adjusted for physical activity, socio-demographic and economic variables, 2017–18.**

| Non-Communicable Diseases | 0dds Ratio | Confidence Interval (95%) | | |
|---|---|---|---|---|
| | | p-value | Lower Bound | Upper Bound |
| **Hypertension** | | | | |
| **Ever smoked/smokeless tobacco use** | | | | |
| **Non smoker** [R] | | | | |
| **Smoker** | 1.39 | 0 | 1.379 | 1.401 |
| **Ever consumed Alcohol** | | | | |
| **Non drinker** [R] | | | | |
| **Drinker** | 1.338 | 0 | 1.327 | 1.349 |
| **Currently Smokes** | | | | |
| **No** [R] | | | | |
| **Yes** | 0.719 | 0 | 0.713 | 0.726 |
| **Currently use tobacco products** | | | | |
| **No** [R] | | | | |
| **Yes** | 0.816 | 0 | 0.809 | 0.822 |
| **Currently consumes alcohol** | | | | |
| **No** [R] | | | | |
| **Yes** | 0.733 | 0 | 0.726 | 0.741 |
| **Diabetes** | | | | |
| **Ever smoked/smokeless tobacco use** | | | | |
| **Non smoker** [R] | | | | |
| **Smoker** | 0.917 | 0.006 | 0.905 | 0.929 |
| **Ever consumed Alcohol** | | | | |
| **Non drinker** [R] | | | | |
| **Drinker** | 2.157 | 0.014 | 2.13 | 2.184 |
| **Currently Smokes** | | | | |
| **No** [R] | | | | |
| **Yes** | 1.087 | 0.008 | 1.071 | 1.103 |
| **Currently use tobacco products** | | | | |
| **No** [R] | | | | |
| **Yes** | 0.963 | 0.006 | 0.951 | 0.975 |
| **Currently consumes alcohol** | | | | |
| **No** [R] | | | | |
| **Yes** | 0.595 | 0.005 | 0.586 | 0.604 |
| **Cancer** | | | | |
| **Ever smoked/smokeless tobacco use** | | | | |
| **Non smoker** [R] | | | | |
| **Smoker** | 19.799 | 0 | 18.819 | 20.83 |
| **Ever consumed Alcohol** | | | | |
| **Non drinker** [R] | | | | |
| **Drinker** | 0.097 | 0 | 0.09 | 0.105 |
| **Currently Smokes** | | | | |
| **No** [R] | | | | |
| **Yes** | 0.246 | 0 | 0.228 | 0.266 |
| **Currently use tobacco products** | | | | |
| **No** [R] | | | | |
| **Yes** | 0.079 | 0 | 0.074 | 0.083 |

*(Continued)*

**Table 3.** (Continued)

| Non-Communicable Diseases | 0dds Ratio | Confidence Interval (95%) | | |
| --- | --- | --- | --- | --- |
| | | p-value | Lower Bound | Upper Bound |
| Currently consumes alcohol | | | | |
| No[R] | | | | |
| Yes | 1.231 | 0.01 | 1.051 | 1.443 |
| Lung Disease | | | | |
| Ever smoked/smokeless tobacco use | | | | |
| Non smoker [R] | | | | |
| Smoker | 2.316 | 0 | 2.28 | 2.352 |
| Ever consumed Alcohol | | | | |
| Non drinker[R] | | | | |
| Drinker | 0.702 | 0 | 0.689 | 0.715 |
| Currently Smokes | | | | |
| No[R] | | | | |
| Yes | 0.532 | 0 | 0.522 | 0.542 |
| Currently use tobacco products | | | | |
| No[R] | | | | |
| Yes | 0.48 | 0 | 0.472 | 0.487 |
| Currently consumes alcohol | | | | |
| No[R] | | | | |
| Yes | 1.665 | 0 | 1.63 | 1.701 |
| Heart Disease | | | | |
| Ever smoked/smokeless tobacco use | | | | |
| Non smoker [R] | | | | |
| Smoker | 2.917 | 0 | 2.866 | 2.969 |
| Ever consumed Alcohol | | | | |
| Non drinker[R] | | | | |
| Drinker | 1.128 | 0 | 1.102 | 1.155 |
| Currently Smokes | | | | |
| No[R] | | | | |
| Yes | 0.258 | 0 | 0.251 | 0.265 |
| Currently use tobacco products | | | | |
| No[R] | | | | |
| Yes | 0.239 | 0 | 0.235 | 0.244 |
| Currently consumes alcohol | | | | |
| No[R] | | | | |
| Yes | 1.87 | 0 | 1.821 | 1.921 |
| High Cholesterol | | | | |
| Ever smoked/smokeless tobacco use | | | | |
| Non smoker [R] | | | | |
| Smoker | 0.701 | 0 | 0.672 | 0.73 |
| Ever consumed Alcohol | | | | |
| Non drinker[R] | | | | |
| Drinker | 0.981 | 0.367 | 0.94 | 1.023 |
| Currently Smokes | | | | |
| No[R] | | | | |
| Yes | 1.331 | 0 | 1.259 | 1.407 |
| Currently use tobacco products | | | | |

(*Continued*)

**Table 3.** (Continued)

| Non-Communicable Diseases | 0dds Ratio | Confidence Interval (95%) | | |
| --- | --- | --- | --- | --- |
| | | p-value | Lower Bound | Upper Bound |
| No[R] | | | | |
| Yes | 0.661 | 0 | 0.631 | 0.693 |
| Currently consumes alcohol | | | | |
| No[R] | | | | |
| Yes | 0.316 | 0 | 0.292 | 0.342 |
| **Bone or Joint Disease** | | | | |
| Ever smoked/smokeless tobacco use | | | | |
| Non smoker [R] | | | | |
| Smoker | 1.095 | 0 | 1.079 | 1.111 |
| Ever consumed Alcohol | | | | |
| Non drinker[R] | | | | |
| Drinker | 1.216 | 0 | 1.197 | 1.236 |
| Currently Smokes | | | | |
| No[R] | | | | |
| Yes | 0.607 | 0 | 0.595 | 0.618 |
| Currently use tobacco products | | | | |
| No[R] | | | | |
| Yes | 0.793 | 0 | 0.781 | 0.804 |
| Currently consumes alcohol | | | | |
| No[R] | | | | |
| Yes | 1.163 | 0 | 1.141 | 1.185 |
| **Stroke** | | | | |
| Ever smoked/smokeless tobacco use | | | | |
| Non smoker [R] | | | | |
| Smoker | 0.574 | 0 | 0.561 | 0.587 |
| Ever consumed Alcohol | | | | |
| Non drinker[R] | | | | |
| Drinker | 1.702 | 0 | 1.666 | 1.739 |
| Currently Smokes | | | | |
| No[R] | | | | |
| Yes | 2.163 | 0 | 2.111 | 2.218 |
| Currently use tobacco products | | | | |
| No[R] | | | | |
| Yes | 0.748 | 0 | 0.73 | 0.766 |
| Currently consumes alcohol | | | | |
| No[R] | | | | |
| Yes | 0.338 | 0 | 0.327 | 0.349 |
| **Neuro/Psychiatric Disease** | | | | |
| Ever smoked/smokeless tobacco use | | | | |
| Non smoker [R] | | | | |
| Smoker | 3.599 | 0 | 3.515 | 3.685 |
| Ever consumed Alcohol | | | | |
| Non drinker[R] | | | | |
| Drinker | 1.408 | 0 | 1.372 | 1.444 |
| Currently Smokes | | | | |
| No[R] | | | | |

(*Continued*)

**Table 3.** (Continued)

| Non-Communicable Diseases | 0dds Ratio | | Confidence Interval (95%) | |
|---|---|---|---|---|
| | | p-value | Lower Bound | Upper Bound |
| Yes | 0.755 | 0 | 0.732 | 0.78 |
| Currently use tobacco products | | | | |
| No[®] | | | | |
| Yes | 0.849 | 0 | 0.83 | 0.869 |
| Currently consumes alcohol | | | | |
| No[®] | | | | |
| Yes | 0.126 | 0 | 0.119 | 0.133 |
| Any NCD | | | | |
| Ever smoked/smokeless tobacco use | | | | |
| Non smoker [®] | | | | |
| Smoker | 1.209 | 0 | 1.203 | 1.215 |
| Ever consumed Alcohol | | | | |
| Non drinker[®] | | | | |
| Drinker | 1.239 | 0 | 1.231 | 1.247 |
| Currently Smokes | | | | |
| No[®] | | | | |
| Yes | 0.683 | 0 | 0.677 | 0.689 |
| Currently use tobacco products | | | | |
| No[®] | | | | |
| Yes | 0.728 | 0 | 0.722 | 0.733 |
| Currently consumes alcohol | | | | |
| No[®] | | | | |
| Yes | 0.59 | 0 | 0.584 | 0.596 |

Here [®]: Reference Category

risk of lung diseases and heart disease respectively as compare with the non-smokers. Chances of having Cancer is 19.8 times more if individual has past smoking behaviour and Neuro and Psychiatric Disease has 3.56 times more that of non-smokers.

Odds of suffering from Any NCD if ever smoked or consumed smokeless tobacco is 1.20 and ever consumed alcohol is 1.23.

Currently smoking individuals have 1.08, 1.33 and 2.16 times higher odds of suffering from Diabetes, High Cholesterol and Stroke respectively and currently consuming alcohol individuals have 1.23, 1.66, 1.87 and 1.16 times more odds of suffering from Cancer, Lung Disease, Heart disease, Bone/Joint Disease respectively.

## Discussion

The results of this study revealed the association between substance use and non-communicable diseases among older people in North-eastern states of India and demonstrated that smoking tobacco, consuming smokeless tobacco, and alcohol consumption were significantly related to poor health in this population. These findings are consistent with earlier research showing that older men and women who do not use any substance have lower rates of all-cause mortality, high blood pressure, diabetes, coronary heart disease, higher levels of cardio-respiratory and muscular fitness (lung diseases), and healthier body mass and composition than older people who use any kind of substance [32].

Findings reveal that the most important correlates of substance use are associated with the MPCE quintile and education i.e., with the increase in economic condition older adults are less likely to use any substance and with the increase in number of years of education people are less likely to use any kind of substance which supports prior studies that males, the uneducated, and the poor had a higher prevalence of tobacco use, which is problematic given that these individuals don't have the necessary resources to take care the morbidity brought on by substance use [30, 33, 34]. From this study results reveal men exhibit a higher likelihood of using nearly all categories of substance compared to women, which also coincides with many literatures [35, 36]; furthermore, this characteristic is more prevalent among men in older ages [37]. SC and STs are less likely to consume substances than other caste groups [38], which shows that there is a significant direct relationship between socioeconomic class and education and alcohol intake in rural tribes. High levels of prevalence were identified in men, age groups of 60 years and above, followed by 30–39 years [39].

Moreover, findings reveal the prevalence of smokeless tobacco to be the highest in the North-eastern region of India which is in support of previous literature [40]. We also found that people residing in rural areas had a slightly higher tendency to use substances compared to the urban areas which is similar to a study [41].

Among all the other non-communicable diseases, hypertension was found to be most prevalent in our study in the North-eastern region [42]. Tripura, Mizoram, Assam, and Manipur were found to have the highest prevalence of using any kind of substance which is above the national average that have been discussed in other studies which may be accounted for the cultural set up in the region [27]. As reported by the United Nations Office on Drugs and Crime (UNODC), drug trafficking across the common border of Myanmar and the North-eastern states of India may have an impact on the increased substance use in these regions [43]. The study found the prevalence of non-communicable diseases in urban areas to be higher compared to rural areas which defends previous studies related to urban-rural differentials in the occurrence of non-communicable diseases [44]. Our study found that older adults who are widowed have a greater likelihood of developing non-communicable diseases compared to those who are currently married. This may be attributed to the fact that many widowed older adults are women, who are more prone to chronic illnesses and may turn to substance use as a way to cope with loneliness and grief [45, 46]. Our study identified significant positive relationships between Lung Disease, Heart Disease, High Cholesterol and smoking behaviour which is in support of previous literature which was conducted in Kerala on the prevalence of coronary heart diseases (CHD), they found high prevalence rates of four risk factors, viz. hypertension, smoking, diabetes, and obesity, where the median age of initiation for both smoking behaviours and alcohol use was 21 years, and nearly two fifths (40%) of the sample population were current smokers and users of alcohol (41%) [47] and also it has been shown earlier that tobacco use and alcohol consumption were the most common risk factors for NCDs among rural tribal adults in Mokokchung district of Nagaland [48]. A study found that the most frequent causes of death in older individuals have been clearly connected to tobacco smoking, which also increases the morbidity and impairment due to several diseases that are prevalent in this age range [49].

Our study also found that with the consumption of alcohol, there is a higher risk of having health problems like Hypertension, Diabetes, Heart Diseases, Bone/Joint Diseases, and Neurological/Psychiatric Diseases, these findings are similar to other studies [50].

Given the growing prevalence of chronic diseases in the aging population of North-eastern states in India, efforts to prevent chronic diseases and improve health behaviours are needed in the region. A step has been initiated by the Government of India for control and prevention

of the highly occurring non-communicable diseases such as Cancer Diabetes, Cardiovascular diseases, and Stroke [51].

## Alcohol and tobacco control policies

Worldwide, there are national alcohol rules in existence in 66 nations. Aside from Bangladesh, Indonesia, and the Maldives, three Muslim nations in Southeast Asia where alcohol drinking is either entirely or partially prohibited in accordance with Islamic belief. The National Authority on Tobacco and Alcohol (NATA) Act of Sri Lanka, which was enacted in 2006, and the Alcohol Beverage Control Act B.E. 2551 of Thailand, which was established in 2008, are the only countries having alcohol policy frameworks in place [52].

In India, numerous tobacco control laws and regulations have been passed in an effort to reduce the incidence such as the Cigarettes Act 1975 (Regulation of Production, Supply, and Distribution), Cigarettes and Other Tobacco Products (Prohibition of Advertisement and Regulation of Trade and Commerce, Production, Supply, and Distribution) Act (COTPA) in 2003, Prevention of Food Adulteration Act (PFA) (Amendment) 1990, Cigarettes and other Tobacco Products (Prohibition of Advertisement and Regulation of Trade and Commerce, Production, Supply, and Distribution) Amendment Rules, 2023, etc. for paan masala and chewing tobacco [53] and for alcohol, the legal drinking age ranges from 18 to 25 years of age, and state-by-state alcohol taxation rates ranging from 30% to 100% on alcoholic beverages are all examples of alcohol control policies. Alcohol is also prohibited in four Indian states and the union territory of Lakshadweep [54].

Additionally, India joined the WHO Framework Convention on Tobacco Control (FCTC) in 2005 as well. In order to manage and control tobacco consumption, WHO introduced MPOWER, for Monitoring tobacco use and prevention policies, protecting people from tobacco smoke, offering help to quit tobacco use, warning about the danger of tobacco, enforcing a ban on tobacco advertising and promotion, and raise taxes on tobacco products. In its National Health Policy published in 2017, the Indian government aims to reduce tobacco consumption by 30 percent on a relative basis by the year 2025.

However, despite everything, substance usage is still pervasive in India [55]. Consequently, it begs important questions like, "Why does it happen this way?" In this context, we attempt in a straightforward manner to analyse the prevalence of NCDs at different levels and their relationship to substance usage among older adults (aged 60+) in the North-eastern states of India. Some major SDG's under which governments are taking steps and suggesting the importance of controlling alcohol and tobacco consumption for the better sustainability of humans are:

SDG 3.a: Strengthen the WHO Convention on Tobacco in all countries.

SDG Target 3.5: Strengthen prevention and treatment of substance abuse.

SDG Target 3.4: By 2030 reduce by one-third pre-mature mortality from non-communicable diseases

Efforts to sensitize people about their consumption behaviour must not be stopped and acts like COTPA must be strengthened and continued, which has important provisions such as: no smoking of tobacco in public places, promotion or advertisement of tobacco product is prohibited, the age limit of 18 years and tobacco products should be sold with graphical warnings on tobacco packets like "Smoking Kills, It causes Cancer". The National Tobacco Control Program was also established in accordance with the provisions of COPTA. It offers a globally coordinated response to the tobacco epidemic and lays out specific guidelines for government

action, such as tax and price measures, bans on tobacco advertising, promotion, and sponsorship, smoke-free zones, etc. It is one of the seven priority areas of NHP, where both preventive and promotive health care is required to achieve sustainable health.

## Limitation of the study

The study focuses on the association between substance use and non-communicable diseases by the different background characteristics using the longitudinal data, but since only 1st wave has been done so far thus it acts as a cross-sectional study and hence no causal relationship has been inferred from the current study. The study is based on the self-reported chronic disease data that may cause the over or under-estimation in the study since there was no metabolic biomarkers data to support risk/association. The study also is based on self-reported consumption of substances which can also hamper our estimations. Moreover, our study was unable to delve into the specific levels of substance consumption behavior. Assessing the risk requires information on the quantity, frequency, and duration of smoking, substance use, and alcohol consumption. Unfortunately, such detailed data is not accessible; only binary responses in the form of "yes" or "no" are available. The study couldn't focus separately on ever-consumed smokeless tobacco and smoking as there was no data, the study also didn't focus on the current consumption of substances in association with non-communicable diseases due to data limitations. Muslim and SC samples are very low as compared to their counterpart in the region, hence it may produce bias in the results. Household consumption which could have been incorporated was not considered because most substance in north northeast region is locally produced and can hamper our study.

## Conclusion and scope of the study

We can conclude that substance use is highly associated with non-communicable diseases in the North-eastern region's older adult population. Our findings reveal much room for improvement in older North-eastern Indians' health behaviours. High tobacco and alcohol use among people aged 60+ with chronic diseases like CVDs, lung diseases, and diabetes is alarming. Prioritizing alcohol reduction, smoking cessation, and physical activity in public health is essential. All over, study shows that people at older age diagnosed with chronic illness are still consuming tobacco and alcohol at high proportion. Few studies have shown that Older adults' not only have higher rate of nicotine dependence but also they are more resistant to quitting [56, 57]. Thus, nothing is impossible as these behaviours are modifiable, hence our findings have important implications for health policy in India. This study will contribute to existing knowledge by examining different factors among older adults aged 60 and we believe it will help us give insight not only into their health associated with their lifestyle behaviours but also it will further help the government implementation of effective policies and programmes for the welfare of older adults.

Efforts to reduce smoking, alcohol, and tobacco consumption among older men are needed like awareness programmes workshops, camps, community discussions, etc. at the community level must be entertained. Also, cultural activities to sensitize the negative impact of substance use may be performed regularly to spread awareness. A complete ban is always not possible as the study region is mainly rural, the substance is locally available, and less expensive as compared with the imported substances. Improved health behaviours are expected to improve health outcomes and prevent or delay the onset of disabilities and chronic diseases. Based on these findings, we recommend the development and implementation of educational and interventional programs to improve health behaviours. These programs may be customized for various groups (e.g., more- and less-educated people, men, and women).

## Author Contributions

**Conceptualization:** Sasanka Boro.

**Data curation:** Sasanka Boro.

**Formal analysis:** Sasanka Boro.

**Methodology:** Sasanka Boro.

**Supervision:** Nandita Saikia.

**Writing – original draft:** Sasanka Boro.

**Writing – review & editing:** Sasanka Boro.

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
