## [Decision Letter · Decision Letter 0]

12 Dec 2023

PONE-D-23-21892The Effects of Substance Use on Non-Communicable Diseases among Older Adults Aged 60 and Above in the North-eastern States of IndiaPLOS ONE

Dear Dr. Boro,

Thank you for submitting your manuscript to PLOS ONE. After careful consideration, we feel that it has merit but does not fully meet PLOS ONE’s publication criteria as it currently stands. Therefore, we invite you to submit a revised version of the manuscript that addresses the points raised during the review process.

We look forward to receiving your revised manuscript.

Kind regards,

Chandan Kumar, Ph.D.

Academic Editor

PLOS ONE

Journal Requirements:

Reviewers' comments:

Reviewer's Responses to Questions

**Comments to the Author**

1. Is the manuscript technically sound, and do the data support the conclusions?

Reviewer #1: Yes

Reviewer #2: Yes

2. Has the statistical analysis been performed appropriately and rigorously? 

Reviewer #1: No

Reviewer #2: Yes

3. Have the authors made all data underlying the findings in their manuscript fully available?

Reviewer #1: Yes

Reviewer #2: Yes

4. Is the manuscript presented in an intelligible fashion and written in standard English?

Reviewer #1: Yes

Reviewer #2: Yes

5. Review Comments to the Author

Reviewer #1: To

The Authors

The Authors prepared the present communication based on the secondary data 'Longitudinal Ageing Study in India (LASI Wave-I, 2017–18)', which was carried out in 35 States/UTs. Why the authors have taken data for only Northeastern states?

The authors looked into the factors responsible for high incidence or prevalence of NCDs including tobacco smoking, substance use, physical inactivity etc.

Methodology

There is need to obtain the data with a specific objective, which is not given in the manuscript. Several disease variable were addressed here, but there no specific definitions for each these variables

In the introduction, it is given like India is a low income country. In fact, India is not low income country. This may be deleted. India is lower middle income country.

How monthly per capita income was collected, which may not be always correct

Quantity, frequency and duration of smoking, substance use, alcohol consumption is very important to estimate the risk, but this kind of data is not available - only except yes or No.

The major drawback in this database, no metabolic biomarkers data to support risk/association

Some of the editorial corrections needed here and there in the text

All the dependent variables collected in the main survey was based on the reported data, which may not be a correct data. At least the data collection on morbidity/mortality should be based on the medical reports/statutory reports

Discussion: May be pruned to precise

The weakness and strengths of the study needs to be mentioned

Conclusions

The conclusions should be focused and should be based on the data

Tables

Table 1 & 2 may be clubbed

Reviewer #2: 1. You can add new tobacco, alcohol, and drugs related policies in the policy section.

2. There has been requirement to highlight the gaps in the former works and say clearly why you are focusing in this field.

3. I think adding the mathematical formulae of Logistic regression would give the study a statistical robustness.

4. In your abstract please edit the results section and explain the abbreviations which are entered in your whole text.

5. Your method part needs to rewrite in extended form.

6. Need to write justification behind the applied statistical analysis. Add justification as of why you have used binary logistic regression.

7. Discussion part need to add some powerful previous works and compare the differences and discuss on them.

8. Add the international classification of disease codes for the diseases that you have selected

9. Literature review and rationale of the need to be improve. Please read this article and cite it in literaure review section (https://doi.org/10.1080/14659891.2022.2146014).

10. How the authors addressed the contribution of other confounding variables in the outcomes of the study.

11. there are several grammatical errors, which need to be fixed before publication.

12. Presentaion of the odds ration should be standard.

13. Please correct it ( line 320, St’s were less likely to consume all kinds)

14. Please try to avoid We, I, you etc. from the manuscript (We found that the most important correlates)

15. I suggests authors to rewrite the discussion section based on key findings only, how its similar or dissimilar with previous study and why?

16. In your study Muslim and SC sample is very low as compared to their counterpart, the results could be biased.

17. A age group specific analyses could be helpful to understand the result better way. It we see the previous results with increasing age the odds of your selected outcome variables is higher than their counterparts. So you could think about it.

6. PLOS authors have the option to publish the peer review history of their article (what does this mean?). If published, this will include your full peer review and any attached files.

Reviewer #1: **Yes: **Avula Laxmaiah, Former Senior Public Health Nutrition scientist, ICMR-National Institute of Nutrition

Reviewer #2: **Yes: **Margubur Rahaman

---

## [Author Response · Author response to Decision Letter 0]

9 Feb 2024

Reviewer #1: To

The Authors

The Authors prepared the present communication based on the secondary data 'Longitudinal Ageing Study in India (LASI Wave-I, 2017–18)', which was carried out in 35 States/UTs. Why the authors have taken data for only Northeastern states?

Response: We selected the north-eastern states based on previous literature. As indicated in the north-east sub-section on page 4, prior research has identified a higher prevalence in this region.

The authors looked into the factors responsible for high incidence or prevalence of NCDs including tobacco smoking, substance use, physical inactivity etc.

Response: Certainly, as outlined in paragraph 3 of the introduction section on page 3, line 121, these factors are acknowledged in the literature as significant contributors to the onset of non-communicable diseases in older age. Therefore, we have duly taken these four elements into consideration

Methodology

There is need to obtain the data with a specific objective, which is not given in the manuscript. 

Response: Certainly, we have added the specific objective at the end of introduction section in the page number 2 line 128.

Several disease variable were addressed here, but there no specific definitions for each these variables

Response: We have added the definition each of our disease(dependent) variable in the dependent variable section of methodology part in the page number 6 line 243.

In the introduction, it is given like India is a low income country. In fact, India is not low income country. This may be deleted. India is lower middle income country.

Response: Certainly, we have corrected the mistake in the conclusion section of the abstract in line 72 of page no. 2. 

How monthly per capita income was collected, which may not be always correct

Response: We have added how MPCE quintile in LASI has been calculated in the socio-economic and demographic variables section of methodology part in the page number 6 line 218. 

Quantity, frequency and duration of smoking, substance use, alcohol consumption is very important to estimate the risk, but this kind of data is not available - only except yes or No.

Response: Yes, we have mentioned this into the limitations section in or study in page number 11 of line 465.

The major drawback in this database, no metabolic biomarkers data to support risk/association

Response: Certainly, we have mentioned this into the limitations section in or study in page number 11 of line 463.

Some of the editorial corrections needed here and there in the text.

Response: We have edited the manuscript and checked for editorial errors.

All the dependent variables collected in the main survey was based on the reported data, which may not be a correct data. At least the data collection on morbidity/mortality should be based on the medical reports/statutory reports

Response: Yes, this problem statement has been discussed in the limitation section in page number 11.

Discussion: May be pruned to precise

Response: Yes, we have edited the discussion section. 

The weakness and strengths of the study needs to be mentioned

Response: Yes, we have given a separate limitation section in page number 11.

Conclusions

The conclusions should be focused and should be based on the data

Response: Yes, the conclusion has been edited and was focused based on our data and findings. 

Tables

Table 1 & 2 may be clubbed

Response: Thank you, the distinction between Table 1 and Table 2 lies in their content: Table 1 provides a description of our study variables, while Table 2 focuses on the prevalence of all Non-Communicable Diseases (NCDs). We made a conscious decision to present them separately to enhance clarity and ensure a more organized presentation of our findings.

Reviewer #2: 

1. You can add new tobacco, alcohol, and drugs related policies in the policy section.

Response: Yes, we have added the COTPA amendment rules in the policy section of page number 10 and line number 419.

2. There has been requirement to highlight the gaps in the former works and say clearly why you are focusing in this field.

Response: Yes, we have mentioned this in the north-east India section page number 5 and line number 191.

3. I think adding the mathematical formulae of Logistic regression would give the study a statistical robustness.

Response: Yes, we have added the mathematical formulae for binary logistic regression in the statistical analysis section of page number 7 in line number 260. 

4. In your abstract please edit the results section and explain the abbreviations which are entered in your whole text.

Response: Yes, we have added the abbreviations in the appendix of the manuscript.

5. Your method part needs to rewrite in extended form.

Response: Certainly, the method section has been revised to provide concise and clear explanations, facilitating a more straightforward understanding of the study.

6. Need to write justification behind the applied statistical analysis. Add justification as of why you have used binary logistic regression.

Response: Yes, we have added the justification behind the usage of binary logistic regression in the statistical analysis section in the page number 6, line number 255.

7. Discussion part need to add some powerful previous works and compare the differences and discuss on them.

Response: Yes, we have added few previous studies to felicitate our findings namely:

1. M. A. Kumar MS, Oinam A, Mukherjee D, Kishore K, “Women who use drugs in NorthEast India,” vol. 25, no. 4, pp. 343–349, 2015, [Online]. Available: https://www.unodc.org/documents/southasia//publications/research-studies/FINAL_REPORT.pdf

2. S. P. Marbaniang, H. S. Chungkham, and H. Lhungdim, “A structured additive modeling of diabetes and hypertension in Northeast India,” PLOS ONE, vol. 17, no. 1, p. e0262560, Jan. 2022, doi: 10.1371/journal.pone.0262560.

3. M. K. Gupta, D. A. Nagdeve, D. Mahata, and H. Chaurasia, “Substance Abuse Among Elderly in India: Evidence Based on Study on Global Ageing and Adult Health (SAGE) Wave 1,” Glob. Soc. Welf., pp. 1–7, 2023

4. M. E. Patrick, P. Wightman, R. F. Schoeni, and J. E. Schulenberg, “Socioeconomic status and substance use among young adults: a comparison across constructs and drugs,” J. Stud. Alcohol Drugs, vol. 73, no. 5, pp. 772–782, 2012

5. R. K. McHugh, V. R. Votaw, D. E. Sugarman, and S. F. Greenfield, “Sex and gender differences in substance use disorders,” Clin. Psychol. Rev., vol. 66, pp. 12–23, 2018.

8. Add the international classification of disease codes for the diseases that you have selected

Response: Certainly, we could have added it but LASI survey does not provide us with the ICD codes hence we would be unable to provide it and hence it has been also added in the limitation section however the variable definition have been added in the dependent variable section of methodology part in the page number 6 and line number 243 for better explanation of the disease variables.

9. Literature review and rationale of the need to be improve. Please read this article and cite it in literaure review section (https://doi.org/10.1080/14659891.2022.2146014).

Response: Thank you for the suggestion, we have added the literature in the north east section to provide rationale to our study in page number 5 in line number 173.

10. How the authors addressed the contribution of other confounding variables in the outcomes of the study.

Response: To adjust for confounding variables, confounding factors such as age, education, marital status, monthly per capita consumption expenditure (MPCE), religion etc. were adjusted in the final model.

11. there are several grammatical errors, which need to be fixed before publication.

Response: Yes, the manuscript has been re-checked for grammatical errors.

12. Presentation of the odds ration should be standard.

 Response: I appreciate your feedback and the attention you've given to the presentation of odds ratios in our paper. Your suggestion to adhere to a standard format is duly noted, and I want to express our gratitude for your valuable input. However, after careful consideration, we have deliberately chosen a specific presentation style for the odds ratios in line with the nuances and objectives of our study. We understand that there are various ways to present odds ratios, and we respect the suggestion for a standard format. Yet, we firmly believe that our chosen approach is the most effective for conveying the nuances of our study.

13. Please correct it ( line 320, St’s were less likely to consume all kinds)

Response: Thank you for your careful attention, we have corrected the typing error.

14. Please try to avoid We, I, you etc. from the manuscript (We found that the most important correlates)

Response: Certainly, we have edited the manuscript and tried to avoid the We and I to the best we can.

15. I suggests authors to rewrite the discussion section based on key findings only, how its similar or dissimilar with previous study and why?

Response: Yes, the discussion has been revised to focus on key findings, incorporating previous literature reviews to highlight similarities and differences with prior studies. 

16. In your study Muslim and SC sample is very low as compared to their counterpart, the results could be biased.

Response: Thank you for the keen observation , the study has been focused in the north-eastern parts of India and according to LASI dataset the sample of Muslim and SC population is low in the region and hence we have added this as one of the limitations in our study. 

17. A age group specific analyses could be helpful to understand the result better way. It we see the previous results with increasing age the odds of your selected outcome variables is higher than their counterparts. So you could think about it.

Response: Thank you for your insightful comment and suggestion regarding age-specific analyses. I appreciate your attention to detail. While I acknowledge the potential value of age-specific analyses in enhancing our understanding, regrettably, conducting such analyses goes beyond the scope of the current paper. Our primary focus was on associating NCDs with substance use among the elderly itself i.e. 60 and beyond. However, I genuinely appreciate your suggestion, and it is duly noted for future research considerations.

---

## [Decision Letter · Decision Letter 1]

3 May 2024

PONE-D-23-21892R1The Effects of Substance Use on Non-Communicable Diseases among Older Adults Aged 60 and Above in the North-eastern States of IndiaPLOS ONE

Dear Dr. Boro,

Thank you for submitting your manuscript to PLOS ONE. After careful consideration, we feel that the revised manuscript still has to undergo a minor modification. Authors are thus advised to go through the comments given in the attached file of the manuscript, and submit the revised version of the manuscript after incorporating the comments/suggestions and undergoing a thorough language editing.

We look forward to receiving your revised manuscript.

Kind regards,

Chandan Kumar, Ph.D.

Academic Editor

PLOS ONE

Journal Requirements:

Reviewers' comments:

**Comments to the Author**

Reviewer #1: All comments have been addressed

2. Is the manuscript technically sound, and do the data support the conclusions?

Reviewer #1: Yes

3. Has the statistical analysis been performed appropriately and rigorously? 

Reviewer #1: Yes

4. Have the authors made all data underlying the findings in their manuscript fully available?

Reviewer #1: (No Response)

5. Is the manuscript presented in an intelligible fashion and written in standard English?

Reviewer #1: Yes

6. Review Comments to the Author

Reviewer #1: Abstract:

Objective: To assess the prevalence of NCDs and strength of association of substance abuse among geriatric population of Northeastern states in India, for the development of preventive strategies, but not for implementation.

Methods:

Data received from the 'Longitudinal Ageing Study in India' (LASI -wave 1 2017-18) for the present communication (paper), not for the study.

Conclusion:

Since, high prevalence of hypertension correlated with the high level of substance abuse, require immediate attention to develop an appropriate intervention strategies for its control (substance abuse) and prevention of hypertension.

7. PLOS authors have the option to publish the peer review history of their article (what does this mean?). If published, this will include your full peer review and any attached files.

Reviewer #1: **Yes: **Avula Laxmaiah, Former Director grade scientist, ICMR-National Institute of Nutrition, Hyderabad, India

---

## [Author Response · Author response to Decision Letter 1]

19 May 2024

The abstract has been changed as advised

---

## [Editor Report · Decision Letter 2]

25 Jun 2024

PONE-D-23-21892R2The Effects of Substance Use on Non-Communicable Diseases among Older Adults aged 60 and above in the North-eastern States of IndiaPLOS ONE

Dear Dr. Boro,

Thank you for submitting your manuscript to PLOS ONE. After careful consideration, we feel that the manuscript has not yet met PLOS ONE’s publication criteria. There is need for a thorough language (English) editing of the manuscript. Please refer to some of the comments, highlighted texts (in yellow colour), and edits (in track change mode) given in the attached copy of your revised manuscript. We invite you to submit a revised version of the manuscript that addresses the points raised in the attached copy of the manuscript.

We look forward to receiving your revised manuscript.

Kind regards,

Chandan Kumar, Ph.D.

Academic Editor

PLOS ONE
---

## [Author Response · Author response to Decision Letter 2]

3 Jul 2024

Yes we have incorporated all the changes as advised by the editor and checked the figures in PACE.

---

## [Editor Report · Decision Letter 3]

9 Jul 2024

The Effects of Substance Use on Non-Communicable Diseases among Older Adults aged 60 and above in the North-eastern States of India

PONE-D-23-21892R3

Dear Dr. Boro,

We’re pleased to inform you that your manuscript has been judged scientifically suitable for publication and will be formally accepted for publication once it meets all outstanding technical requirements.

Kind regards,

Chandan Kumar, Ph.D.

Academic Editor

PLOS ONE

---

## [Editor Report · Acceptance letter]

30 Aug 2024

PONE-D-23-21892R3 

PLOS ONE

Dear Dr. Boro, 

I'm pleased to inform you that your manuscript has been deemed suitable for publication in PLOS ONE. Congratulations! Your manuscript is now being handed over to our production team.

Kind regards, 

on behalf of

Dr. Chandan Kumar 

Academic Editor

PLOS ONE